

# Intolerance of uncertainty and conditioned place preference in opioid addiction

Milen L. Radell[1], Michael Todd Allen[2], Belinda Favaloro[3], Catherine E. Myers[4,5], Paul Haber[6], Kirsten Morley[6] and Ahmed A. Moustafa[3,7]

[1] Department of Psychology, Niagara University, Lewiston, NY, United States of America
[2] School of Psychological Sciences, University of Northern Colorado, Greeley, CO, United States of America
[3] School of Social Sciences and Psychology, Western Sydney University, Sydney, New South Wales, Australia
[4] Department of Veterans Affairs New Jersey Health Care System, East Orange, NJ, United States of America
[5] Department of Pharmacology, Physiology & Neuroscience, New Jersey Medical School, Rutgers University, Newark, NJ, United States of America
[6] Discipline of Addiction Medicine, Central Clinical School, The University of Sydney, Sydney, New South Wales, Australia
[7] Marcs Institute for Brain, Behaviour and Development, Western Sydney University, Sydney, New South Wales, Australia

Corresponding author
Ahmed A. Moustafa,
a.moustafa@westernsydney.edu.au

## ABSTRACT

Several personality factors have been implicated in vulnerability to addiction by impacting learning and decision making. One such factor is intolerance of uncertainty (IU), the tendency to perceive uncertain situations negatively and avoid them. Conditioned place preference (CPP), which compares preference for contexts paired with reward, has been used to examine the motivation for both drug and non-drug rewards. However, preference for locations associated with non-drug reward, as well as the potential influence of IU, has not been thoroughly studied in individuals with addiction. In the current study, we examined CPP using a computer-based task in a sample of addicted individuals undergoing opioid maintenance treatment and never-addicted controls. Patients were confirmed to have higher IU than controls. In the CPP task, the two groups did not differ in overall time spent in the previously-rewarded context. However, controls were more likely than patients to immediately return to this context. Contrary to our predictions, IU was not a significant predictor of preference for the previously-rewarded context, although higher IU in controls was associated with a higher number of rewards obtained in the task. No such relationship was found in patients.

## INTRODUCTION

Drug addiction is thought to involve a disruption of reward processing such that the pursuit and use of a substance come to dominate an individual's behavior, at the expense of other actions, despite the associated negative consequences (*Wise & Bozarth, 1987*; *Clark & Robbins, 2002*). A combination of positive and negative reinforcement processes may contribute to initial substance use, with prolonged exposure leading to changes in brain systems involved in motivation that mark the transition to addiction (*Wise & Bozarth, 1987*; *Everitt & Robbins, 2005*; *Everitt et al., 2008*). However, not all individuals exposed to drugs develop addiction, suggesting that some are more vulnerable than others. As one

example, the personality trait of intolerance of uncertainty (IU) involves the tendency to perceive uncertain situations as aversive, and can result in avoidance and negative expectations about the consequences of such situations (*Nelson et al., 2016*). IU is higher in individuals with opioid use disorder (OUD), raising the possibility that it is a risk factor for this, and potentially other, substance abuse disorders (*Carleton, Norton & Asmundson, 2007*; *Garami et al., 2017*).

Individuals with higher IU appear to require more information before making decisions, presumably, to reduce uncertainty. For example, in one study, participants with higher IU sampled more marbles compared to those with lower IU before estimating the number of black vs. white marbles in a bag (*Ladouceur et al., 1998*). In addition, those with higher IU appear to prefer low-probability immediate rewards over high-probability but delayed rewards, suggesting that these individuals find waiting under uncertainty aversive, and are willing to take risks (e.g., choose a low-probability reward) as long as doing so allows them to avoid this aversive state (*Luhmann, Ishida & Hajcak, 2011*). Taken together, these results suggest that IU can bias decision-making under uncertain conditions, including those involving probabilistic rewards.

There is a large body of evidence that decision-making is altered in addiction. For example, individuals with substance abuse disorders tend to choose immediate small rewards over larger but delayed rewards, and also show a preference for immediate certain rewards over larger, but uncertain rewards (*Clark & Robbins, 2002*; *MacKillop et al., 2011*). One factor that could influence decision-making is reward sensitivity, which appears to be decreased in addiction, especially for non-drug rewards or cues that signal these rewards (*Blum et al., 2000*; *Koob et al., 2004*; *Volkow et al., 2010*; *Luijten et al., 2017*). This decrease, associated with a loss of motivation for non-drug rewards, could bias decision-making in favor of continuing drug use, which has become the dominant source of reinforcement. It may also contribute to relapse after a period of abstinence, which poses a major hurdle for the treatment of substance use disorders (*Goudriaan et al., 2008*).

Conditioned place preference (CPP) is a common paradigm that can be used to assess reward sensitivity in terms of preference for contexts previously paired with reward. This task is widely used in the study of addiction in animal models in order to understand the reinforcing effects of both drugs and other types of rewards, including food and access to a mate (*Bardo & Bevins, 2000*; *Tzschentke, 2007*). Several computer-based analogues have been developed for use in humans (*Childs & De Wit, 2009*; *Molet, Billiet & Bardo, 2013*; *Astur, Carew & Deaton, 2014*; *Childs, Astur & De Wit, 2017*). Most relevant to the current work is a study that examined whether IU was associated with CPP in a probabilistic task where college students could forage for reward (i.e., points in the task) in two virtual rooms, one of which was associated with more frequent reward (*Radell et al., 2016*). This study found that, when given a choice, participants with higher IU tended to return to the room previously associated with a higher chance of reward, while those with lower IU showed no such preference (*Radell et al., 2016*). There are multiple possible interpretations of this preference, including that it may have resulted from a tendency to choose the more certain choice, i.e., returning to a location known to contain reward in the past, or from an increased tendency to pursue reward (*Radell et al., 2016*).

While IU is predicted to have both cognitive and behavioral effects, most studies have focused on its cognitive aspects (*Luhmann, Ishida & Hajcak, 2011*). As discussed above, although research has started to investigate the relationship between IU and reward sensitivity, to our knowledge, none have done so in a population with OUD. Thus, the current study adopted the CPP task of *Radell et al. (2016)* to compare the potential behavioral effects of IU between individuals undergoing treatment for OUD and healthy controls. First, consistent with prior studies (*Carleton, Norton & Asmundson, 2007*; *Garami et al., 2017*), we hypothesized that patients would report higher IU than controls. Second, we hypothesized that if patients have reduced reward sensitivity to non-drug reward, they would show reduced preference for the context associated with reward relative to controls. Third, we expected IU to modulate behavior in both groups, i.e., within both the patient and control groups, high-IU individuals would have higher preference for the rewarded context than low-IU peers.

## MATERIALS AND METHODS

### Participants

Participants included outpatients of the Royal Prince Alfred Hospital (RPAH), Sydney, Australia, with OUD undergoing drug maintenance treatment at the RPAH Methadone Clinic. The control group consisted of healthy volunteers without a history of drug addiction or other psychiatric disorders. The final sample included 27 controls and 32 patients with detailed demographic and medication information presented in the results. All participants gave written informed consent before the initiation of any procedures. This study was approved by the Research Ethics and Governance Office at the Royal Prince Alfred Hospital (approval number: HREC/12/RPAH/295) and was carried out in accordance with guidelines established by the Human Research Ethics Committees.

### Procedure

For patients, testing took place in a quiet room at the RPAH Methadone Clinic within 30 min of receiving the daily drug dose. Controls were tested in a quiet room at the Western Sydney University or, as needed, in their homes. All participants completed the Intolerance of Uncertainty Scale (IUS) (*Buhr & Dugas, 2002*) and a computer-based CPP task adopted from a previous study (*Radell et al., 2016*), which was administered on a MacBook Pro. The CPP task, illustrated in Fig. 1, consisted of a tutorial, pre-training, training and a post-training phase. Participants controlled a cartoon avatar (a fox) and were instructed to help the fox collect as many golden eggs as possible; exact instructions are provided in the Appendix.

During the tutorial phase, the fox was placed in a lobby area with a single door. Participants were told that they could click on the door to enter a room ("tutorial room"). The tutorial room was visually distinct from the other rooms subsequently encountered in the task. The tutorial room contained eight chests located in a circular pattern and participants were required to click on the chests until they found two golden eggs. As a participant clicked on a chest, the fox moved to examine the chest, which opened and revealed whether there was an egg inside or not. During the tutorial, every chest contained
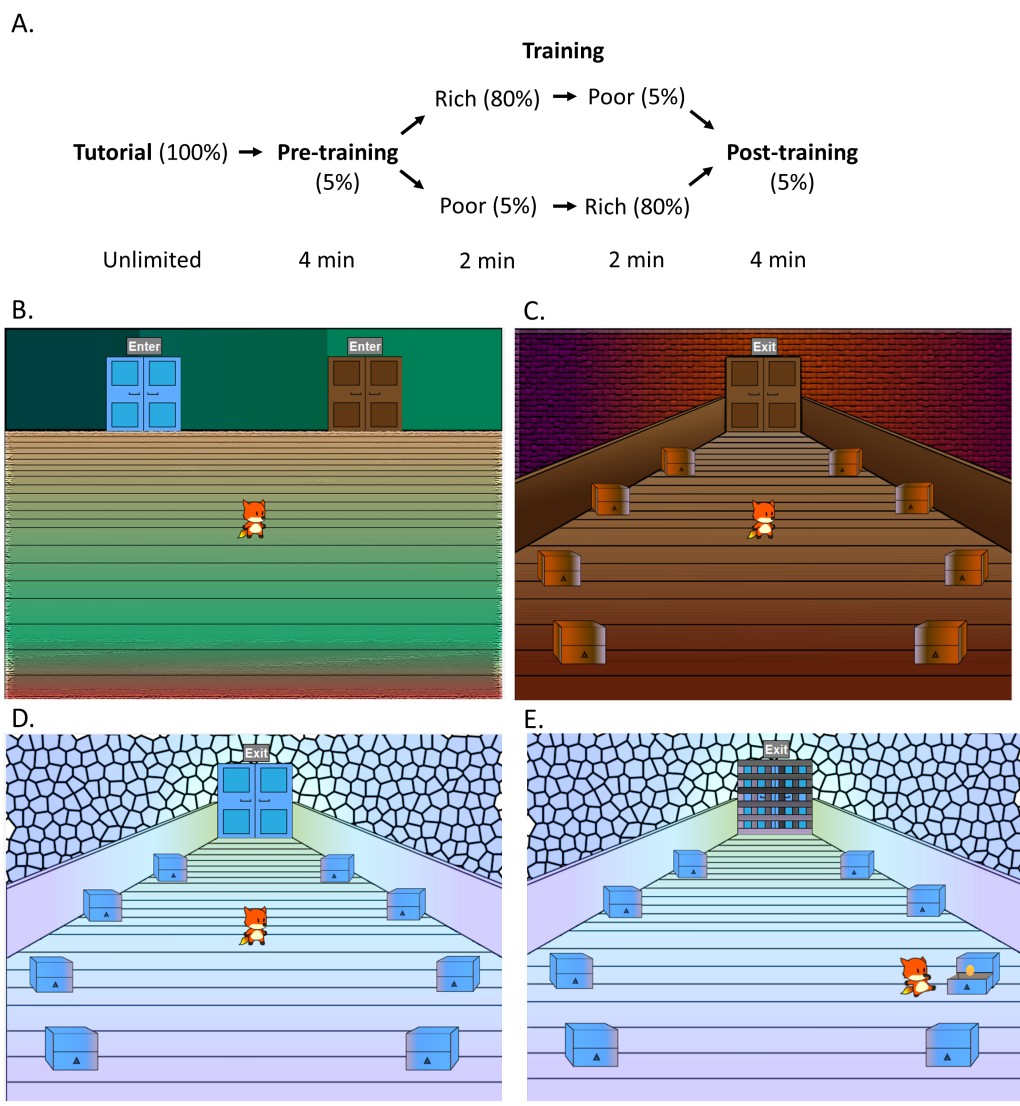

**Figure 1** **Design of the conditioned place preference task.** (A) The task consisted of a tutorial, pre-training, training and a post-training phase. (B) Participants controlled an avatar (the fox), which was placed in the lobby area (shown here) at the start of each phase. The lobby area contained two doors, with door order counterbalanced. During the pre-training and post-training, participants could switch between a (C) brown and a (D) blue room by using the mouse to click on the doors, and could also click on the chests to search for golden eggs, increasing their total score. Each chest initially had a 5% chance of containing an egg. (E) In the training phase, participants could only enter one room at a time, with order counterbalanced, and were locked inside. In one room ("rich room"), each chest initially had an 80% chance of containing an egg, in contrast to the other ("poor room") where each chest initially had a 5% chance. Figure adapted from *Radell et al. (2016)*.

an egg and therefore, the first two distinct chests the participants clicked completed this phase. The total number of eggs collected was always visible at the top of the screen but was reset at the conclusion of the tutorial phase, and therefore the two eggs collected found during the tutorial were not included in the final count.

Next, in the pre-training phase, the fox was returned to the lobby with the option of two doors (blue and brown) to choose from. The left or right placement of the doors was counterbalanced across participants. The doors led to two visually distinct rooms (blue and brown), which each contained eight chests to explore. During this phase, which lasted 4 min, participants were free to explore both of the rooms and click on the chests to search for eggs. Participants had a 5% chance of obtaining an egg from any chest, and once they had gained an egg, the chance to be rewarded again from the same chest decreased to 0 and then increased by 1% every 4 sec. Participants were not notified of these reward contingencies and therefore had to rely on trial and error. The time spent in each room, the total number of chests searched, and the total number of eggs collected in the task, were recorded.

Following the pre-training was a training phase. During the training phase, the room where participants had spent more time during the pre-training (i.e., the more-preferred room) was assigned to be "poor" while the less-preferred room was assigned to be "rich." In the "rich" room, each chest initially had a 0.8 probability of containing an egg; in the "poor" room, each chest initially had a 0.05 probability of containing an egg. Once an egg was collected from a chest, the probability of an egg from the same chest decreased to 0 and then increased by increments of 0.1 every 4 sec. Training consisted of two parts lasting 2 min each. In each part, the fox was placed in a lobby with one door unlocked and the other locked; the participant could then enter the unlocked room and spend the duration collecting eggs. The order in which rooms were unlocked (rich vs. poor in part 1 vs. part 2) was counterbalanced across participants. In the second part of training, the locks were switched, so that the participant could now spend the duration of part 2 collecting eggs in the other room. The order in which subjects searched chests, total number of chests searched and total score of eggs was again recorded.

The final phase was the post-training, which was identical to the pre-training. The fox was placed in the lobby with both rooms unlocked. Reward contingencies for this phase also followed the same pattern as the pre-training. In this phase, the first room that the participant entered was also recorded.

## Data analysis

Demographic variables were compared between groups using $t$-test for continuous variables (e.g., age) and chi-square test for categorical variables (e.g., gender distribution).

IU scores were compared between patient and control groups using independent-samples $t$-test. For all analyses that examined the influence of IU on behavior, participants were classified as high or low on IU based on a cut-point equal to the sample median score on the IUS.

Analysis of behavior in the CPP task first considered immediate preference, defined as the first room entered during the post-training phase (rich or poor). Since this factor is categorical, three-way hierarchical loglinear analysis was employed. Loglinear analysis is an extension of the chi-square test for three or more categorical variables. Thus, the factors included in the model were the first room entered in the post-training phase (rich vs. poor), group (control vs. patient) and IU (high vs. low). A similar analysis was performed

to examine whether the last room experienced during the training phase (rich vs. poor), was related to the first room entered in the post-training phase.

Second, we examined the overall preference for the rich room, defined as the percent of time which a participant spent in the rich room, computed as (time in rich/(time in rich + time in poor)) * 100, separately for the pre- and post-training phase. A mixed-model analysis of variance (ANOVA) was used, with a within-subjects factor of phase (pre-training vs. post-training phase) and between-subjects factors of group (patient vs. control) and IU (high vs. low). Note that since the rich room was defined for each participant based on pre-training performance (i.e., the less preferred room at pre-training became the rich room in training) the overall preference pre-training was always ≤50%.

The tendency to explore (i.e., switch between rooms) was also examined because overall preference as defined above might result from a reduced tendency to explore (such that more time is spent in a particular room) or alternatively, a tendency to switch between rooms, but immediately revert back to a particular room. Exploration was defined as the total number of room entries (with rich and poor room entries added together), computed separately for the pre- and post-training phase. The total number of mouse clicks on chests, again irrespective of the specific room, was also considered as a measure of foraging behavior. Mixed-model ANOVAs were performed on the average total number of room entries and chest clicks, with a within-subjects factor of phase (pre-training vs. post-training) and between-subjects factors of group (control vs. patient) and IU (low vs. high).

Finally, as a general measure of task performance, a between-subjects ANOVA was performed on the total score obtained in the task, with factors of group (patient vs. control) and IU (high vs. low), in order to examine whether a particular group of participants was more successful at foraging for reward. The total score was computed by summing the total number of rewards (i.e., golden eggs) obtained by participants across the pre-training, training and post-training phases. The tutorial phase (which always yielded two eggs) was excluded from this computation.

As appropriate, Levene's test and Mauchly's test were used to confirm equality of variance. Where Levene's test indicated significant deviations from the assumption of normality, Welch's $t$-test was used. Where Mauchly's test indicated significant deviations from equality of variance ($p < 0.05$), Greenhouse-Geisser correction was used to adjust degrees of freedom for interpreting $p$-values from $F$-values. Where multiple comparisons were made, Bonferroni correction was used to protect against accumulating risk of Type I error. It is important to note that two-tailed tests were employed in all statistical analyses since we had no prior data from this population, and it was possible that groups could differ in either direction.

## RESULTS

### Demographics and personality variables

A total of eight participants (four patients and four controls) were excluded due to computer failure. The demographic data for the final sample ($n = 59$) are shown in Table 1. There were significantly more female participants in the control than in the patient group,

**Table 1 Demographic and personality data.** Education was unknown or could not be determined for 16 patients and one control and was only calculated for the remaining sample.

|  | Controls | Patients |
|---|---|---|
| Number of participants | 27 (19 female) | 32 (6 female) |
| Age range (years) | 20–65 ($M = 39.74$, $SD = 13.98$) | 25–61 ($M = 43.13$, $SD = 9.82$) |
| Education range (years) | 8–18 ($M = 13.04$, $SD = 3.01$) | 9–16 ($M = 11.63$, $SD = 2.31$) |
| Mean score on IUS | 65.85 ($SD = 24.32$) | 82.59 ($SD = 27.03$) |

confirmed by a Pearson chi-square test, $\chi^2(1) = 15.980$, $p < 0.001$. Thus, in the current sample, group was confounded with gender. A Welch's $t$-test showed that there were no significant differences in age between controls and patients, $t(45.553) = 1.057$, $p = 0.296$. The age at which drug abuse started was reported by 23 patients and ranged from 13 to 39 ($M = 21.04$, $SD = 6.30$). Out of the patients, 26 were on methadone maintenance (mean dose $= 103.83$ mg/day, $SD = 52.01$, range 30–200) and six received buprenorphine (mean dose $= 20$ mg/day, $SD = 8.29$, range 6–30). For five of the six, this was a combination of buprenorphine and naloxone (Suboxone). Based on medical records, most patients also had comorbid psychopathology, most commonly including depression, bipolar disorder, anxiety, post-traumatic stress disorder and schizophrenia.

Scores on the IUS ranged from 30 to 135 ($M = 74.93$, $SD = 26.95$). A higher score on this scale indicates greater intolerance of uncertainty. As expected, OUD patients had significantly higher IU than controls, $t(57) = 2.48$, $p = 0.016$, $r = 0.31$ (Table 1). For subsequent analyses, participants were split into high and low IU groups based on the sample median of 78, with 31 individuals (13 patients, 18 controls) classified as low and 28 (19 patients, nine controls) classified as high (i.e., had a total score >78 on the IUS). A Pearson's chi-square test indicated no significant differences in gender distribution between individuals with high and low IU, $\chi^2(1) = 0.005$, $p = 0.943$. A Welch's $t$-test confirmed controls and patients also did not differ in education, $t(37.59) = 1.699$, $p = 0.098$.

## Conditioned place preference task
### Immediate preference
The first analysis examined whether there was an immediate preference to enter the previously-rich room at the start of the post-training phase. Since the control group was mostly female, while the patient group was mostly male, a Pearson chi-square test was used to confirm that there was no significant preference to enter the rich room first as a function of gender, $\chi^2(1) = 0.519$, $p = 0.471$, eliminating this as a possible confound. Next, a hierarchical three-way loglinear analysis was performed with factors of the first room entered in the post-training (rich vs. poor), IU (high vs. low) and group (control vs. patient). This produced a model that retained only the two-way interactions with a likelihood ratio of $\chi^2(3) = 9.092$, $p = 0.028$. The three-way interaction and main effects did not make a significant contribution to the model, with likelihood ratios of $\chi^2(1) = 2.016$, $p = 0.156$ and $\chi^2(3) = 4.433$, $p = 0.218$, respectively. The two-way interaction between the first room entered and group was significant, $\chi_p^2(1) = 4.964$, $p = 0.026$, indicating that controls were significantly more likely than patients to enter the rich room first (Fig. 2A). Based on

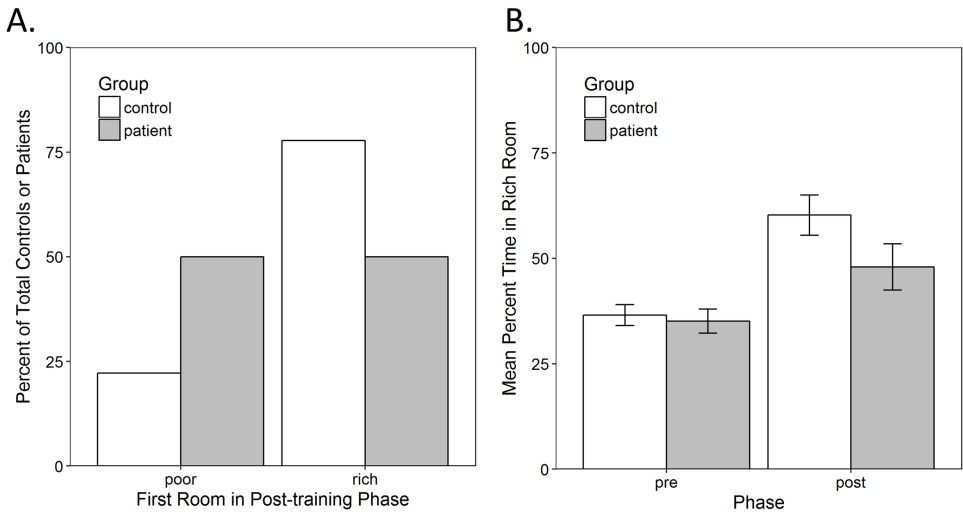

**Figure 2  Conditioned place preference in the task.** (A) Immediate preference shown as the percent of total controls or patients who entered the previously-poor or -rich room first at the start of the post-training. There was a significant interaction between group and the first room entered ($p = 0.026$). Most controls entered the rich room first, while an exactly equal number of patients entered the rich and the poor rooms. (B) Overall preference shown as the average percent of the time spent in the rich room during the pre- and post-training phase. There was a significant increase in the time spent in the rich room from pre- to post-training ($p < 0.001$) but no other significant main effects or interactions. Error bars represent standard error of the mean (SEM).

the odds ratio, the odds of a control entering the rich room first were 3.5 (1.12, 10.96) times higher than a patient entering this room. The two-way interaction between IU and group was also significant, $\chi_p^2(1) = 4.031, p = 0.045$. The odds of a patient with high IU were 2.92 (1.01, 8.49) times higher than those of a control. This was not surprising since patients had higher IU than controls and so there were more patients in the high IU group. However, the two-way interaction between first room entered and IU was not significant, $\chi_p^2(1) = 0.085, p = 0.771$.

Since the first room entered in the post-training may also be a function of which room was experienced during the second training phase, an additional hierarchical three-way loglinear analysis was conducted to examine this potential confound, with factors of the first room entered (rich vs. poor), the room experienced in the second training phase (rich vs. poor) and group (control vs. patient). This produced a model that retained only the two-way interactions, with a likelihood ratio of $\chi^2(3) = 12.833, p = 0.005$. The three-way interaction and main effects did not make a significant contribution to the model, with likelihood ratios of $\chi^2(1) = 0.389, p = 0.533$ and of $\chi^2(3) = 5.658, p = 0.129$, respectively. The two-way interaction between the first room entered and the last room experienced was significant, $\chi_p^2(1) = 7.776, p = 0.005$. There were 4.14 (1.35, 12.66) higher odds of a participant entering the rich room first if they were in the poor room during the second training phase. The two-way interaction between the first room entered and group was once again significant, $\chi^2(1) = 6.207, p = 0.013$, confirming that controls were more likely to enter the rich room than patients. The two-way interaction between group and the last

room experienced was not significant, $\chi^2(1) = 1.325, p = 0.250$, therefore, training order was successfully counterbalanced and the last room experienced cannot account for the increased tendency of controls to enter the rich room.

*Overall preference*

Next, to examine whether participants tended to spend more time in the rich room overall, a mixed-model analysis of variance (ANOVA) was performed, as described in the methods, on the percent time spent in the rich room during the pre-training and post-training phase of the task. There was a significant main effect of phase, $F(1, 55) = 17.136, p < 0.001$, $\eta_p^2 = 0.238$, indicating that participants increased the amount of time spent in the rich room from pre-training to the post-training (Fig. 2B). There were, however, no other significant main effects or interactions.

*Exploration*

As described in the methods, a mixed-model ANOVA was performed on the total number of room entries to assess exploration, with a within-subjects factor of phase (pre-training vs. post-training) and between-subjects factors of group (control vs. patient) and IU (low vs. high). There was only a significant main effect of test, $F(1, 55) = 15.988, p < 0.001, \eta_p^2 = 0.225$, indicating that number of room entries decreased from the pre-training to the post-training (Fig. 3A). There were no other significant main effects or interactions (all $p > 0.17$).

A mixed-model ANOVA was also performed on the total number of mouse clicks on chests with a within-subjects factor of room (rich vs. poor) and between-subjects factors of group (control vs. patient) and IU (low vs. high). There was a significant main effect of group $F(1, 55) = 7.853, p = 0.007, \eta_p^2 = 0.125$, and a group $\times$ IU interaction, $F(1, 55) = 5.019, p = 0.029, \eta_p^2 = 0.084$. There were no other significant main effects or interactions (all $p > 0.22$). As expected, the interaction indicated that controls with high IU searched more chests compared to high IU patients while low IU controls and patients searched a similar number of chests (Fig. 3B). Bonferroni-corrected independent-samples $t$-tests (alpha adjusted to $0.05/2 = 0.025$) confirmed a significant difference in the total number of chests clicked by high IU controls compared to high IU patients, $t(26) = 3.238$, $p = 0.003, r = 0.54$. There was no difference between low IU controls and patients, $t(29) = 0.440, p = 0.663$.

*Total score*

Since high IU control participants searched significantly more chests, this raises the possibility that they also obtained a higher total score in the task (i.e., found more golden eggs). To confirm this, a between-subjects ANOVA was performed on the total score in the task computed as described in the methods, with factors of group (patient vs. control) and IU (high vs. low). There was a significant IU by group interaction, $F(1, 55) = 11.811, p = 0.001, \eta_p^2 = 0.177$, indicating that individuals with low IU obtained a similar total score regardless of group, while among those with high IU, controls found more eggs than patients (Fig. 4). Bonferroni-corrected independent-samples $t$-tests (alpha adjusted to $0.05/4 = 0.0125$) found a significant difference between controls and patients with high IU, $t(26) = 4.881, p < 0.001, r = 0.69$, but not between controls and

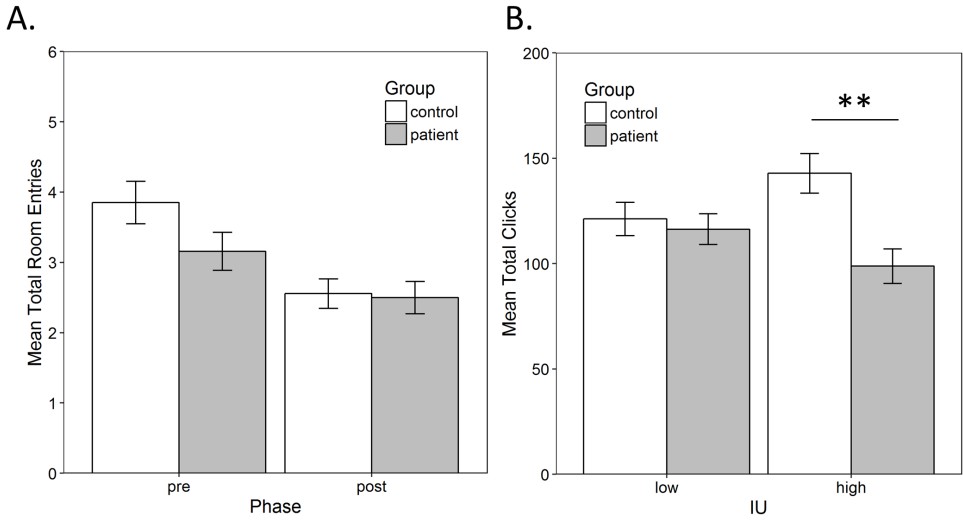

**Figure 3 Exploration.** (A) The average total number of room entries during the pre- and post-training phase. There was a significant decrease in room entries from pre- to post-training ($p < 0.001$), which was not related to group. (B) The average total mouse clicks on chests. Controls with high IU searched significantly more chests compared to high IU patients ($p < 0.001$) while low IU controls and patients searched a similar number of chests ($p = 0.663$). Error bars represent SEM. **, Significant at $p < 0.001$.

patients with low IU, $t(29) = 0.100, p = 0.921$. Among controls, those with high IU had a significantly higher score than those with low IU, $t(25) = 2.891, p = 0.008, r = 0.501$, but there was no difference in the total score obtained by patients with low and high IU, $t(30) = 1.771, p = 0.087$. There was also a significant main effect of group, $F(1, 55) = 10.856, p = 0.002, \eta_p^2 = 0.165$, with controls collecting more eggs than patients but, as shown by the interaction, this was driven by controls with high IU. Finally, the main effect of IU was not significant ($p = 0.206$). A similar analysis was performed by replacing group with gender to rule this out as a potential confound given that more patients were male, while more controls were female. The gender × IU interaction failed to reach significance, $F(1, 55) = 3.179, p = 0.080, \eta_p^2 = 0.055$. The main effects of gender and IU were also not significant (both $p > 0.10$).

## DISCUSSION

The results show that the training phase resulted in an immediate preference for the previously-rich room in controls but not in patients, who instead entered both rooms at approximately equal rates at the start of post-training (Fig. 2A). A possible explanation for this difference in preference is that chronic drug use in the OUD patient group may have altered reward thresholds and decreased reward sensitivity (*Volkow et al., 2010*). For example, chronic drug use downregulates dopamine receptors and dopamine production, providing a possible explanation for why addicted individuals have lower sensitivity to natural rewards, such as food, sex and water (*Volkow et al., 2002*). Thus, it is possible that the OUD patients had lower sensitivity or motivation for the rewards used in the current task. However, it is important to note that, across all participants, there was an increase in the

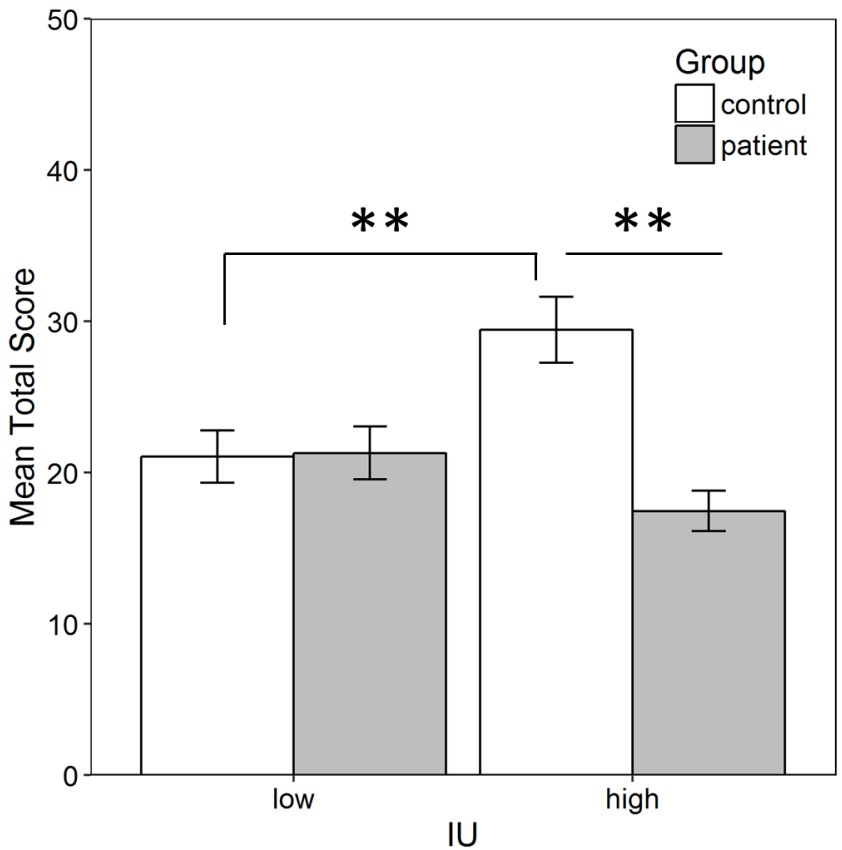

**Figure 4** **Average total score in the task.** Total score was calculated as the sum of the points (i.e., number of golden eggs) acquired by participants across each phase of the task, excluding the two points always obtained during the tutorial. Controls with high IU obtained a significantly higher score than both patients with high IU ($p < 0.001$) and controls with low IU ($p < 0.001$). Error bars represent SEM. **, Significant at $p < 0.001$.

time spent in the previously-rich room from pre-training to post-training, which suggests that both groups were sensitive to reward contingencies (Fig. 2B). Alternatively, since the poor room was always assigned to be the more-preferred room based on the pre-training phase, the tendency of patients to enter the poor room could also reflect preference to revisit a previously-preferred location, even if that location was not associated with reward.

Another possible explanation is that a different search strategy, such as foraging, was used by patients. Foraging involves choosing where and how to look for reward, including weighing a number of options associated with different reward contingencies (*Platt & Huettel, 2008*; *Kolling et al., 2012*). Typically, there is an element of risk or cost associated with leaving a location to search in a new location. This may include time and energy lost traveling to this location as well as having to assess the value of the possible rewards. Since there was little risk of time or energy loss in the current task, it is possible that patients may have more readily adopted a strategy of moving between rooms after obtaining reward (i.e., adopted a win-shift strategy). Drug addicts may spend most of their time foraging,

especially under the control of stimuli that act as reinforcers (*Belin et al., 2013*). This is also consistent with an increase in reward seeking following expectancy violations, as previously reported in individuals with OUD (*Myers et al., 2016*). In the current task, this would have occurred during the post-training phase when the probability of reward decreased to pre-training levels. This behavior is not necessarily pathological, though it may not be optimal in situations where reward can only be found in a specific location.

However, in the current study, both patients and controls appeared to switch between rooms to a similar extent, and across participants, there was a decrease in this type of exploratory behavior from pre-training to post-training (Fig. 3A). This is likely the result of conditioning during the training phase, since both controls and patients showed an increase in the amount of time they spent in the rich room, from pre-training to post-training (Fig. 2B). Thus, the decrease in exploration is consistent with the observed increase in preference for the rich room. In the future, the task could be modified to examine preference when there is greater risk involved. For example, both reward and punishment learning could be assessed by introducing a chance to lose points when foraging in particular locations (e.g., high risk and high reward vs. low risk and low reward). This may affect preference and the strategy used by participants.

As expected, patients were also found to have greater IU than controls (Table 1), which is consistent with past research (*Carleton, Norton & Asmundson, 2007*; *Nelson et al., 2016*; *Garami et al., 2017*). However, neither of the two indicators of place preference (i.e., the first room entered and the room participants spent more time in during the post-training) was related to IU. This was contrary to our initial hypothesis that individuals, and in particular, controls with high IU should be more likely to enter the previously-rich room first during the post-training, as found in an earlier study using the same task (*Radell et al., 2016*). Instead, while more controls than patients did enter the rich room first, in the current study this was not related to IU. The prior study also found no increase in the time spent in the rich room from the pre-training to the post-training phase, again contrary to the current results where training increased the amount of time participants spent in the previously-rich room. Possible explanations for these discrepant findings include differences in the populations studied. The study of *Radell et al. (2016)* was conducted on college students who were considerably younger (mean age 20.7 years) and more highly educated than the controls in the current study. On the other hand, the patients in the current study had been subject to repetitive drug use over a number of years, and were currently on opioid maintenance, which may have impacted learning, motivation and decision-making.

Surprisingly, while IU was not related to preference, it was associated with overall performance in the task. Specifically, the results show that controls with high IU collected more eggs than both patients with high IU and controls with low IU (Fig. 4). This was largely due to searching more chests (Fig. 3B). In contrast, in a previous study with this task, there were no differences in total number of eggs collected by participants with high vs. low IU (*Radell et al., 2016*). There are a number of possible explanations for this tendency, including differences in motivation to acquire reward or in motor activity. It is also consistent with previous studies that have shown individuals with high IU require

more information before making decisions under uncertain conditions (*Ladouceur, Talbot & Dugas, 1997*; *Ladouceur et al., 1998*). In this case, the decision may have been to continue searching or to give up or slow down, since it was not possible to leave the room during training. Thus, high IU controls may have continued to search in order to more clearly discern reward contingencies. If this was the case, however, similar results should have been obtained in high-IU patients as in high-IU controls. Instead, high-IU patients did not search as many chests and, therefore, did not collect as many eggs.

Although several studies have examined CPP in humans, to our knowledge, none have done so in a population undergoing treatment for OUD and using secondary reinforcers (e.g., points in a computer game). For example, in a study by *Childs & De Wit (2009)*, humans received d-amphetamine or placebo in separate rooms, with the results indicating a preference for the drug-paired room (*Childs & De Wit, 2009*). In *Molet, Billiet & Bardo (2013)*, a distinct virtual environment was paired with either pleasant music or static noise. Participants spent more time in the environment paired with pleasant music (*Molet, Billiet & Bardo, 2013*). *Astur, Carew & Deaton (2014)* also examined preference for two virtual rooms, one paired with chocolate, finding that participants spent more time in the chocolate-paired room, but only if they were food deprived (*Astur, Carew & Deaton, 2014*). *Astur et al. (2016)* assessed preference in a similar paradigm but with a secondary reinforcer (i.e., points) in young adults and found they spent more time in the points-paired room, and rated it as more enjoyable (*Astur et al., 2016*). Finally, *Childs, Astur & De Wit (2017)*, assessed preference for a virtual room paired with a higher vs. lower monetary reward. The results showed a transient preference, both in terms of time spent in each room and subjective liking for the room with the higher reward, which was no longer apparent 24 hours later (*Childs, Astur & De Wit, 2017*). Similarly, in the current study, there was an increase in the time spent in a room with a higher probability of reward over one with a much lower probability, although this preference did not appear to differ between OUD patients and controls. Future studies could examine whether the type of reward used would modulate preference in addicted vs. non-addicted groups.

The current task differed from the typical non-human animal version of the CPP paradigm (for review, see *Bardo & Bevins, 2000*) in several ways. First, in the current task, it was not sufficient to simply be present in a particular room to obtain reinforcement, as in animal studies. Rather, participants were required to actively forage different locations for reward within the room. This may help increase the external validity of the task since reward seeking (e.g., for monetary, or other types of rewards) often involves active responding rather than simply going to and remaining at a particular location. Second, in animal tasks reward is usually delivered with 100% contingency in the rewarded room. In the current task, reward was probabilistic in that not every chest contained an egg. This was done to increase the task's sensitivity to the potential effect of IU on behavior. Third, in animal studies, pre- and post-training tests are usually conducted in the absence of reward. In the current task, reward could be obtained in both the pre- and post-training phases, although the probability was low (5% for both the rich and poor rooms). This was to ensure participants remained motivated and to reduce frustration, although this opens the possibility that the overall preference for some participants was influenced by any rewards

discovered. However, this effect is expected to be small given the relatively low chance of reward and limited time available during the post-training phase. Fourth, animal studies typically involve multiple conditioning sessions, spread over several days, with training and testing on separate days. The current study employed short, one-time 2 min training sessions, with a 4-min pre-training and post-training phase. Despite this, we still detected a significant increase in the time spent in the rich room from pre- to post-training (Fig. 2B) suggesting that training was successful, though it might be possible to amplify this effect by extending the duration of training.

## Limitations

A limitation of the study was that, in the current sample, most of the patients were male while most of the controls were female, making gender a possible confound in the interpretation of the results. However, it is important to note that while group was found to be a significant predictor of immediate preference, gender was not. Due to the small inclusion of female patients, it was also not possible to examine potential gender differences in the OUD group. Gender differences in opioid abuse and dependence are well documented (*Back et al., 2010*; *Back et al., 2011a*; *Back et al., 2011b*). For example, in a large sample from the National Survey on Drug Use and Health, *Back et al. (2010)* found greater self-reported lifetime and past year use and abuse of opioids in men compared to women. In addition, while both men and women have low rates of treatment utilization, this is even lower in women (*Back et al., 2010*) and may lead to a greater proportion of men entering methadone maintenance treatment, as sometimes reported (*Chatham et al., 1999*; *Puigdollers et al., 2004*). Methadone maintenance treatment outcomes also differ between males and females (*Bawor et al., 2015*).

There may also be gender differences in individual reasons for drug use. Opioid use in men appears more likely to be motivated by a positive reinforcement mechanism (e.g., recreation), while in women it may be motivated by a negative reinforcement mechanism (e.g., self-medication) (*Back et al., 2010*; *Back et al., 2011a*). Gender differences in opioid abuse also depend on a number of other demographic factors, including age, education, and employment (*Back et al., 2011a*; *Bawor et al., 2015*). In the current sample, patients and controls did not significantly differ in age or education. However, education could not be reliably determined for a large number of the patients, and future studies should include a measure of intelligence rather than rely on self-report. In addition, as in the current sample, most individuals who use or abuse opioids also report other drug use, with women more likely than men to take drugs that can enhance opioid effects (*Fischer, Cruz & Rehm, 2006*; *Back et al., 2011a*). Thus, the observed differences between patients and controls cannot definitively be attributed exclusively to the effects of opioid use. Nonetheless, it can be argued the current sample is more representative of the OUD population by including polydrug users.

Patients were also tested within 30 min of receiving their daily maintenance drug dose, which could mean that differences between the groups, at least in part, reflect acute opioid effects rather than long-term changes in reward processing resulting from chronic drug abuse. Future studies could test patients on maintenance therapy later in the day, or right

before drug administration. However, both methadone and buprenorphine have relatively long elimination half-lives with large individual differences, e.g., for methadone, values ranging from less than 15 hours to over 60 hours have been reported (*Eap, Buclin & Baumann, 2002*; *Elkader & Sproule, 2005*). This long half-life contributes to making these drugs particularly suitable for substitution therapy, but also implies that individuals must be tested after a period of abstinence to completely eliminate drug effects as an alternative explanation.

## CONCLUSIONS

In summary, while patients undergoing opioid maintenance treatment were confirmed to have higher IU than controls, no relationship was found between IU and preference for a room previously paired with a high probability of reward. However, IU did predict the total number of rewards obtained in the task—high IU controls outperformed low IU controls, as well as high and low IU patients. Controls, irrespective of IU, were more likely to enter the rich room than patients. This preference may reflect differences in motivation or learning about reward between the two groups, including changes in how contextual cues that predict secondary reinforcers are processed. It may also, at least in part, reflect acute opioid effects, since patients had received opioid treatment just prior to participating in the study. Future research should examine these possibilities further since understanding the factors that govern decision-making and learning processes in addicted individuals can contribute to developing or improving current treatment options, including the prevention of relapse.

## ACKNOWLEDGEMENTS

The opinions expressed herein are those of the authors and do not necessarily represent the views of the Department of Veterans Affairs or the U.S. Government.

## APPENDIX

Instructions and prompts displayed on the computer screen over the course of the computer-based conditioned place preference task.

The first screen seen by participants:

*Help Kit the Fox collect golden eggs. First, you'll practice how to collect eggs.*

*When you're ready to begin, click start.*

This starts the tutorial. The following is displayed at the bottom of the screen:

*Try moving between rooms. Click on the center door.*

After clicking on the door:

*Great! Now, click on the chests to search for eggs. Try to find 2 golden eggs.*

Whenever an egg is found:

*+1*

The score counter displayed at the top also increments. When 2 eggs are found, the following is displayed:

*Next, you'll start the real game.*
*Try to collect as many eggs as you can before time runs out.*
*When you're ready to begin, click start.*
This starts the pre-training phase. When time in each phase runs out:
*End of round <round number>. Please wait…*
At the end of each phase, except for the post-training, the screen fades out and the avatar appears back in the lobby. At the end of the post-training, the following appears on the screen:
*Thank you for playing.*
*Please tell the experimenter you're done.*

### Funding
The authors received no funding for this work.

### Competing Interests
Catherine Myers is an Academic Editor for PeerJ. The other authors declare that there are no competing interests.

### Author Contributions
- Milen L. Radell conceived and designed the experiments, analyzed the data, contributed reagents/materials/analysis tools, prepared figures and/or tables, authored or reviewed drafts of the paper, approved the final draft.
- Michael Todd Allen and Catherine E. Myers conceived and designed the experiments, authored or reviewed drafts of the paper, approved the final draft.
- Belinda Favaloro conceived and designed the experiments, performed the experiments, authored or reviewed drafts of the paper, approved the final draft.
- Paul Haber and Kirsten Morley conceived and designed the experiments, contributed reagents/materials/analysis tools, authored or reviewed drafts of the paper, approved the final draft.
- Ahmed A Moustafa conceived and designed the experiments, performed the experiments, analyzed the data, contributed reagents/materials/analysis tools, authored or reviewed drafts of the paper, approved the final draft.

### Human Ethics
The following information was supplied relating to ethical approvals (i.e., approving body and any reference numbers):

This study was approved by the Research Ethics and Governance Office at the Royal Prince Alfred Hospital and was carried out in accordance with guidelines established by the Human Research Ethics Committees. HREC/12/RPAH/295.

### Data Availability
The raw data are included as Supplemental Files.

## Supplemental Information

Supplemental information for this article can be found online at http://dx.doi.org/10.7717/peerj.4775#supplemental-information.

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
