# Peer review of "Intolerance of uncertainty and conditioned place preference in opioid addiction"

_PeerJ, doi:10.7717/peerj.4775_

## Round 0.1 · original submission · Major Revisions

I have now received the comments from reviewers and while they both agree on the article being well written and well presented, they would like to see further revisions of the Introduction (reorganisation and tighetening) and clarification and justification of the hypothesis and statistical analyses. I also concur with one reviewer that a discussion contrasting and comparing the animal and human CPP would strengthen the paper.

Reviewer 1 ·

Basic reporting

Basically well done. Some comments are provided in the feedback below

Experimental design

Basically fine. Some clarifications on the statistics and data computation are needed

Validity of the findings

Some discussion as to the validity of the model are needed

Additional comments

The paper by Radell et al is well-written and attempts to adopt a preclinical paradigm to the study of personality variables in substance use disorder. In general the paper is well laid out and well presented. Some points of clarification are below.
My main comment is on the use of this novel paradigm in humans. Although it is interesting to adapt a preclinical model for study in humans, further discussion of the parallels between the preclinical model and human model is needed. Further, some discussion of the validity of this model is needed. For example, in the current model, the final phase involves exploration of the two rooms with a reward contingency in place. In the animal model, the final exploration phase is in the absence of any reward. This is important because the authors talk about ‘preference’, but is it really a preference if a reward is available? What are some of the strengths and weaknesses of the differences between the preclinical and clinical model?
In the Methods section it would be helpful if the authors could provide a section describing the statistical analyses, with a rationale for the statistics chosen. This is important because the statistics were different for different parts of the study. Is it not possible to analyze all the data with ANOVAs? In this section it is also necessary to describe the computation of all variables, such as in the final phase. What exactly was a ‘higher score’ (e.g. line 266) and how was this computed? Why did the authors analyze the tendency to switch between rooms, and how was this computed?
Throughout the paper the authors describe the patients as ‘addicted’. Would it not be more accurate to simply state that they have opiate use disorder (OUD)? (or substance use disorder (SUD)?)
1) Line 44: Please define a motivational hierarchy
2) Line 57: a citation is needed
3) The paragraph on line 72 beginning: ‘IU has also been associated…’. Is this paragraph necessary?
4) Paragraph on line 82: Is there evidence for changes in reward sensitivity from non-imaging data? That would seem more relevant to the present discussion, would it not?
5) Line 183: ‘Questionnaires’ should read ‘Demographics and Personality Variables’ to match the header on the Table
6) Line 196: there should be some mention as to what high IUS scores mean, so the reader can follow along easier with this logic.
7) line 315: why does this behavior decrease from pre-training to post-training? Some interpretation is warranted. Also, why was this analysis done? There is no rationale in the Methods
8) Line 382: A citation is needed.
9) Figures: Error bars should be included for all figures. Statistics should also be presented in the figure captions, with symbols on the figures representing the results of pairwise comparisons.
10) Fig 2A: Is the difference between the groups in the poor room significant? They look to be.
11) Fig 3A figure caption: Mention of IU is made, but there is nothing about IU in the figure.

·

Basic reporting

Concerning the basic reporting, the paper is well-written, well documented (i.e., solid list of references) ; the structure and the editing appear to be conform to PeerJ standards as I understand them. Figures were helpful to « visualize » the characteristics of the CPP task. Finally raw data was supplied. My main concern lies with the background to show context of the current study. The text from line 42 to line97 is quite disconnected for the goal of the study, the findings reported elsewhere and the theoretical aspects do not nicely revolve around the goal of the current study that is clearly stated from line 98 to line 125. I suggest that the authors reorganize the text before that with the goal in mind to set up a better context for their study.

Experimental design

The quality of both experimental design and the tested ideas are good and they dovetail nicely with the scope of the journal. The research question that consists in looking at how people with a drug addiction behave in a CPP preparation to understand their sensitivity to reward distribution has of some merits though the study has some limitations. But those limitations are clearly presented in the text and they do not preclude later publication. My major concern is that the authors predict both scenarios for the results if I am not mistaken. To paraphrase them : « Second, it was also hypothesized that if patients have reduced reward sensitivity to non-drug reward, they would show reduced preference for the context associated with reward relative to controls. However, IU may also modulate preference given its prior association with decreased reward sensitivity in individuals with depression (Nelson, Shankman & Proudfit, 2014) but increased tendency to return to a previously-rewarded context in healthy young adults (Radell et al., 2016). » Say differently on a formal plan with two alternative A and B, they seem to predict A but also B. I found this approach questionable because one may think that more « thinking » would have be necessary to make a specific prediction. This needs more elaboration to understand the reason of this choice. At the present moment, the set of hypotheses require bilateral tests. As a friendly reminder, the reject of the hypothesis is decided whether the test value is significantly different, even it is inferior (link reject area) or superior (right reject area). The test is said unilateral when the null hypothesis checks if a value is superior or equal to the test value (link unilateral) or inferior or equal to this value (right unilateral). When the null hypothesis consists in testing the equality of the test value with a given value, the test is bilateral.

Validity of the findings

Validity of the findings is good. Especially because both methods and statistical analyses seem robust. My only concern here is that some size effects are missing (e.g., Chi square tests).

Additional comments

"no comment'

---

## Round 0.2 · accepted · Accept

The authors have conducted a thorough revision and have greatly improved the manuscript.

# Reviewer 1 ·

Basic reporting

No comment

Experimental design

No comment

Validity of the findings

No comment

Additional comments

The authors has addressed all concerns. Thank you for the detailed explanations and for the considered revisions.

·

Basic reporting

Good

Experimental design

Good

Validity of the findings

Good

Additional comments

The implemented changes greatly contributed to improving the final version of the paper.